# A REP-FAMSEC Method as a Tool in Explaining Reaction Mechanisms: A Nucleophilic Substitution of 2-Phenylquinoxaline as a DFT Case Study

**DOI:** 10.3390/molecules26061570

**Published:** 2021-03-12

**Authors:** Brian Kamogelo Mdhluli, Winston Nxumalo, Ignacy Cukrowski

**Affiliations:** 1Department of Chemistry, Faculty of Natural and Agricultural Sciences, University of Pretoria, Lynnwood Road, Hatfield, Pretoria 0002, South Africa; u15106129@tuks.co.za; 2Department of Chemistry, Faculty of Science and Agriculture, University of Limpopo, Private Bag X1106, Sovenga 0727, South Africa

**Keywords:** REP-FAMSEC, ONSH, quinoxaline, nucleophilic addition, hydrolysis

## Abstract

In search for the cause leading to low reaction yields, each step along the reaction energy profile computed for the assumed oxidative nucleophilic substitution of hydrogen (ONSH) reaction between 2-phenylquinoxaline and lithium phenylacetylide was modelled computationally. Intermolecular and intramolecular interaction energies and their changes between consecutive steps of ONSH were quantified for molecular fragments playing leading roles in driving the reaction to completion. This revealed that the two reactants have a strong affinity for each other, driven by the strong attractive interactions between Li and two N-atoms, leading to four possible reaction pathways (RP-C2, RP-C3, RP-C5, and RP-C10). Four comparable in energy and stabilizing molecular system adducts were formed, each well prepared for the subsequent formation of a C–C bond at either one of the four identified sites. However, as the reaction proceeded through the TS to form the intermediates (**5a**–**d**), very high energy barriers were observed for RP-C5 and RP-C10. The data obtained at the nucleophilic addition stage indicated that RP-C3 was both kinetically and thermodynamically favored over RP-C2. However, the energy barriers observed at this stage were very comparable for both RPs, indicating that they both can progress to form intermediates **5a** and **5b**. Interestingly, the phenyl substituent (Ph1) on the quinoxaline guided the nucleophile towards both RP-C2 and RP-C3, indicating that the preferred RP cannot be attributed to the steric hindrance caused by Ph1. Upon the introduction of H_2_O to the system, both RPs were nearly spontaneous towards their respective hydrolysis products (**8a** and **8b**), although only **8b** can proceed to the final oxidation stage of the ONSH reaction mechanism. The results suggest that RP-C2 competes with RP-C3, which may lead to a possible mixture of their respective products. Furthermore, an alternative, viable, and irreversible reaction path was discovered for the RP-C2 that might lead to substantial waste. Finally, the modified experimental protocol is suggested to increase the yield of the desired product.

## 1. Introduction

Quinoxaline is a chemical compound that is made up of benzene and pyrazine rings fused together. It is characterized as a bioisostere of naphthalene, quinoline, and benzothiophene [1]. In recent year, these heterocyclic compounds have attracted a lot of attention in medicinal chemistry as they have been identified as pharmacologically important compounds due to their distinct biological properties. Quinoxaline derivatives are active against bacteria, fungi, leishmania, tuberculosis, malaria, depression, cancer and neurological activities, etc., [2]. These compounds are also of great interest due to their potential in fighting pathophysiological conditions such as Alzheimer’s diseases and epilepsy [3]. Therefore, quinoxaline derivatives are regarded as an important class of *N*-heterocyclic compounds in organic synthesis and drug discovery. Not only is quinoxaline important in medicine but this substructure serves as a skeleton for the design of many heterocyclic compounds that are important in fluorescent dyeing agents, electroluminescent materials, chemical switches, and semiconductors [4,5,6,7,8,9].

The reaction of quinoxaline derivatives with aryl and alkyl nucleophiles has been investigated extensively and mechanisms have been proposed [10,11,12,13,14,15,16]. This reaction is assumed to follow the oxidative nucleophilic substitution of hydrogen (ONSH) reaction mechanism shown in Scheme 1. This addition-elimination mechanism is characterized by three steps, namely *nucleophilic addition*, *hydrolysis,* and *oxidation* [17,18]. It was reported in the experimental work that the nucleophilic substitution in **1** occurs at position 3, as shown in Scheme 1. According to classical organic chemistry, position 3 is the preferred site for nucleophilic substitution since it retains the aromaticity of the molecule post the oxidation step of the reaction.

It is commonly accepted that the mechanism shown in Scheme 1 is well understood. However, following the general and applicable protocols, Ndlovu et al. [16] reported yields of widely varying degrees when 2-substitued quinoxalines reacted with different carbon-based nucleophiles. Not entirely surprisingly, the steric hindrance was proposed as the contributing factor affecting the yields of these reactions. However, no other additional study, either experimental or theoretical, was conducted to substantiate this claim. As a matter of fact and in contrast to the quinoxaline derivatives vast medicinal and other applications, there are no reports on reaction mechanisms from computational modelling.

Hence, with an aim of explaining the large variation in experimental yields, we decided to explore the reaction mechanism computationally. To achieve that, we made use of a reaction energy profile and fragment attributed molecular system energy change (REP-FAMSEC) protocol, which was recently reported [19]. This general-purpose protocol provides qualitative and quantitative information on the primary forces driving or preventing the reaction to proceed to completion. Atoms and molecular fragments playing a significant role in the reaction are identified and their energetic contribution is monitored along potential reaction pathways. As a case study, the reaction of 2-phenylquinxaline (**1**) and lithium phenylacetylide (**2**) to give 2-phenyl-3-(2-phenylethynyl)quinoxaline was selected.

## 2. Computational Details

All DFT calculations were performed in Gaussian 16 version B01 [20] at the B3LYP [21]/6-311++G(d,p) [22] level of theory with Grimme’s D3 empirical dispersion correction (GD3) [23,24]. The solvent effects were simulated using the standard polarizable continuum model (PCM) [25] with tetrahydrofuran (THF) as the selected solvent. The vibrational frequency analysis was used to confirm stationary points and transition states (TSs) along the reaction paths. As expected, none and one negative (imaginary) frequency were found, respectively. To confirm that the identified TSs join the relevant reactants and the expected intermediates or product, intrinsic reaction coordinate (IRC) calculations were performed [26]. The conformational search (by varying the torsion angle of every rotatable bond in the system by 30°) for reactants and adducts was performed in Spartan using the molecular mechanic force field (MMFF) and the Monte Carlo algorithm [27]. The AIMAll [28] software was used to generate molecular graphs and the quantum theory of atoms in molecules (QTAIM) [29] / interacting quantum atoms (IQA) [30] data required for computing energy terms were implemented in the REP-FAMSEC method [19]. The XYZ coordinates of all structures considered (Appendix A) and a full set of their energies (Appendix A) are included in Appendix A.

## 3. Results and Discussion

Figure 1 shows the molecular graphs of **1** and **2** with the QTAIM-defined net atomic charges (*Q*(A)). It is instantly apparent that the Li-atom of **2**, with *Q*(Li) of +0.9463*e*, must be attracted to the N-atoms most, as their *Q*(N) are more negative than −1*e* (*Q*(N1) = −1.1164*e* and *Q*(N4) = −1.1211*e*). Notably, these atoms carry the largest +/− atomic charges. Therefore, the interaction between Li and N atoms must be seen as the driving force that brings **1** and **2** together.

The molecular graph of **1** also reveals that there are four positively charged C-atoms, namely C2, C3, C5, and C10. Importantly, they are the only C-atoms in **1** with positive net charges: +0.553 ± 0.013*e* for C2 and C3, +0.435 ± 0.013*e* for C5 and C10. C-atoms of this nature (positively charged) are called electrophilic sites, and this means that, in principle, there are four places where a C–C bond formation can take place. Despite this, there are only two possible approaches that **2** can make towards **1**. That is, **2** must swing either towards C2 or C10 when Li interacts with N1 or swing towards C3 or C5 when Li interacts with N4.

From an eye inspection of **1**, one might argue that steric hindrance (if playing any role at all) should be more significant for the C2 and C3 substitution sites due to the presence of the phenyl substituent at C2. However, from a charge perspective, C2 and C3 carry charges that are more positive than C5 and C10 thus making them more susceptible for nucleophilic addition. It is clearly difficult to make any conclusive findings based on the atomic charges and assumed steric hindrance. Therefore, to gain further insight, we will analyze each step on the reaction energy profiles (REPs; see Figure 2) computed for the entire process. We will start from the adduct formation (**3**), through the nucleophilic addition step computed for the C2, C3, C5, and C10 substitution sites (TS **4** and intermediate **5**), followed by the spontaneous adduct formation between **5** and the H_2_O molecule (**6**), and finally, the hydrolysis step with its TS **7** and product **8**. Furthermore, we will explore an unexpected highly competing reaction pathway (RP) discovered by us for the C2 substitution site (it is marked as C2 (C16) in Figure 2).

### 3.1. Nucleophilic Addition

#### 3.1.1. Adduct Formation (3)

Upon the formation of an adduct (**3** in Figure 2), no bonds are broken or formed within the separate molecules (see Figure 3). However, changes in their molecular geometries are expected, structures and structural features of the energy-optimized molecules from **1** to **5** along all four reaction pathways can be found in Appendix A. As mentioned, the organolithium nucleophile, **2**, can approach **1** in two different ways forming adducts **3a**, **3b, 3c,** and **3d** that are, in principle, well prepared for the subsequent formation of a C–C bond at the C2, C3, C5, and C10 electrophilic site, respectively.

A full set of energies obtained on the adduct formation (the electronic energy (*E*), zero-point vibrational corrected energy (*E*_ZPVE_) [31,32], enthalpy (*H*), and the Gibbs free energy (*G*)) of the reactants (**1** and **2**), as well as the four adducts, is included in Appendix A. Notably, a trend **3b** < **3c** ≈ **3d** < **3a** holds for all energy terms obtained. This is somewhat surprising, as C2 appears to be the least likely substitution site even though net atomic charges of C5 and C10 are significantly less positive. However, very comparable changes in all energy terms (*E*_ZPVE_, *H,* and *G*) computed for all adducts (**3a**–**d**) are seen in Figure 2. This strongly suggests that all adducts are likely to form and hence, there is no hindrance at this stage of the reaction.

In any reaction consisting of two (or even more) distinct molecules, attractive intermolecular interactions drive the molecules towards each other. Hence, formation of adduct or complexes takes place. Importantly, the same interactions that drive the adduct formation will also be major players leading to a transition state and an intermediate formation. To gain an insight on the relative stability of adducts **3a**–**d** and what drives their formation and hence the entire reaction, we first analyzed the diatomic molecular fragments involved in the strongest attractive interactions along all RPs (see Table 1). A full set of the most attractive and repulsive diatomic interactions in **3**, **4,** and **5** along the four reaction pathways considered is included in Appendix A. From this data, it is seen that indeed this is the Li27-atom that interacts the strongest with the respective N-atom of **1** along all the RPs as we predicted from the net atomic charges-based analysis shown in Figure 1.

Considering molecule **2**, lithium phenylacetylide (Figure 1), the interaction energy between Li bonded to C2 was computed to be −106.9 kcal mol^−1^ which was mainly (85%) of classic coulombic nature ($V_{\mathrm{cl}}^{\mathrm{Li}1,C2}$ = −90.9 kcal mol^−1^). That is, only 15% of this interaction was of covalent character ($V_{\mathrm{XC}}^{\mathrm{Li}1,C2}$ = −16.0 kcal mol^−1^). Importantly, we discovered that the strength of this interaction had not only weakened across all RPs upon the formation of adducts (see the second entry for the interaction energies between Li and C28 in Table 1) but is much weaker than the Li ···N interactions. To this effect, the interaction of −185 kcal mol^−1^ between Li and N4 of **1** (this leads to the substitution on C3) is twice as strong as the interaction between the same Li-atom and C28 of **2**. Therefore, one can argue whether this is indeed an adduct formation reaction or is it a spontaneous formation of a first and stable intermediate. We also noted that the XC-term contributed only 5% to the leading Li···N4 interaction and three times more (15%) to the Li···C28 interaction on the **3b** adduct formation. Therefore, it is clear that the nature of the Li···C28 interaction stays the same and is of more covalent character than the Li···N4 interaction. Classically, this might be interpreted as there is no definite shift of the Li-atom from **2** to **1**. Hence, this is indeed an adduct formation reaction. One must note, however, that the interaction between Li and N4 atoms is by far more ionic due to the much larger charge difference when compared with the Li,C28 atom-pair and this explains the much smaller covalent contribution. Notably, the considerations equally apply to all four hypothetical reaction pathways.

Lastly, it is important to point on a geometric aspect that a classical chemist would definitely consider. The inter-nuclei distance d(Li,C2) of 2.04 Å in **2** elongated to d(Li,C28) of 2.06 Å in **3b**. At the same time, d(Li,N4) in **3b** is 2.15 Å, hence slightly longer (by 0.09 Å) than the internuclear distance between Li and C28. Regardless and importantly, these inter-nuclear distances are significantly shorter than the sum of the Van Der Waals radii of the respective atoms [33]. This strongly suggests that the electron cloud of Li overlaps with the electron clouds of both C28 and N4 in **3b**. As such, the adduct **3b** might be then seen as a new molecule with the Li-atom acting as a “linker” connecting two organic moieties. In general, this is quite an interesting case where classical interpretations of bonding are failing a chemist. Considering the above observations and the role played by Li-atom throughout the synthetic process, we decided the most convenient way forward is to treat the interactions between Li and all other atoms of **1** and **2** as of intermolecular character.

Data in Table 1 reveal that there are other atom-pairs involved in very strong, either attractive or repulsive, interactions and they must be accounted for. Therefore, these atoms were grouped to form molecular fragments of special interest in this work, namely:All atoms of the quinoxaline moiety in **1**, 𝒬 = {N1-H15}.All atoms of the phenyl substituent at C2, 𝒫𝒽1 = {C16-H26}.Two molecular fragments made from atoms of 𝒬, namely (a) all atoms of the benzene ring, ℬ𝓃 = {C5-C10,H12-H15}, and (b) all atoms of the pyrazine ring, 𝒫 = {N1-C5,C10,H11}.Two molecular fragments made from atoms of 𝒫. The first is made of highly negatively charged N-atoms, 𝓝 = {N1,N4}, and the second containing highly positively charged C-atoms, 𝓒 = {C2,C3,C5,C10}.Three-atom molecular fragment of **2**, 𝓛 = {Li27,C28,C29}.Two molecular fragments made from atoms of 𝓛, one with a highly positively charged Li-atom and another with highly negatively charged C-atoms, 𝓐 = {C28,C29}.All atoms of **2** except Li27, 𝓡 = {C28-H40}.All atoms of the phenyl ring in **2** 𝒫𝒽2 = {C30-H40}.

The following molecular fragments were also considered to account for a 3-atom and 4-atom environment for the on-coming nucleophile:(a)𝒢1 = {C2,N1,C10} and 𝒢2 = {C3,N4,C5}.(b)ℱ1 = {N1,C2,C3,C16}, ℱ2 = {C2,C3,N4,H11}, ℱ3 = {N4,C5,C6,C10}, and ℱ4 = {C5,C10,C9,N1}.

The computed relevant interfragment interaction energies are summarized in Table 2. A full set of data obtained for all reaction pathways considered on the adduct formation is included in Appendix A.

Focusing on the interactions between the largest molecular fragments (entries 1–8 in Table 2), one can make several important observations:(a)Except entry 8, the interaction energies computed for the RP-C3 are most favorable (i.e., they are most negative) and this is in agreement with the lowest energy barrier seen for this pathway in Figure 2, as well as with experimental data.(b)Notably, regardless of the approach between the two molecules, **1** and **2** have a strong affinity for each other as highly negative values are observed for entry 1. This large intermolecular attraction is mainly and by far due to the huge affinity between **1** and the Li-atom. This is clearly recovered by about 40 kcal mol^−1^ weaker attraction between **1** and fragment 𝓡 (see also entry 11 showing the interaction energy between **1** and Li).(c)Quite unexpectedly, the 𝒫𝒽1 fragment of **1** plays an important role by guiding **2** toward RP-C2 or RP-C3 (see entries 3–5). The computed attractive interactions are in direct contrast to the classical organic chemistry as one would expect the phenyl substituent at C2 to sterically hinder the nucleophilic addition along RP-C2 and RP-C3, more especially for RP-C2. It is then clear that the preferred RP cannot be attributed to the steric hindrance linked with or caused by 𝒫𝒽1. Attractive interactions between 𝒫𝒽1 and 𝒫𝒽2 (entry 5) also explain the loss of linearity in **2** upon the formation of adducts **3a** and **3b**.(d)Entry 8 shows the repulsive interaction between the ℬ𝓃 fragment of **1** and the entire molecule **2**. This holds for all RPs and makes C5 and C10 highly unfavorable substitution sites in accord with large energy barriers TS **4**, as seen in Figure 2.

In order to provide a clear picture of the impact made by the Li-atom, two molecular fragments, namely 𝓛 which is made of Li, C28 and C29, and 𝓐, which differs from 𝓛 by the absence of Li, were considered. Starting with the interactions of the individual atoms of 𝓛 in **1** (Li, C28 or C29) with the entire molecule **1**, we found that they are attractive and very comparable across all RPs, particularly for Li and C28 (see entries 11–13). However, these atoms combined interactions as the molecular fragment 𝓛 with the entire molecule **1** are in favor of RP-C3. Surprisingly, this is mainly due to the contribution made by C29 (see entry 13), as the interactions between Li and **1** are highly comparable for all RPs (see entry 11). More interactions between either 𝓛 or 𝓐 and a specific molecular fragment of **1** are included in Appendix A. From this discussion, it follows that although the Li-atom interacts strongest with **1**, it only takes up the responsibility of driving **2** into the vicinity of **1**. That is, Li27 or any other single atom does not differentiate between the RPs.

We have established that Li is attracted the strongest to **1** through its interactions with the N-atoms (Table 1). It is then reasonable to assume that the strong attractive interactions between 𝒫 and **2** (entry 7 in Table 2) must be a result of the combined interactions between the combined (i) attractive interactions of the highly negatively charged N-atoms (as a molecular fragment 𝓝, entry 14) and (ii) repulsive interactions of the positively charged C-atoms (as a molecular fragment 𝓒, entry 15).

Recalling that the N-atoms are the docking space for the Li-atom which plays the important role of guiding the two reactants towards each other, the interactions of the immediate three-atom environment (fragment) around the N-atoms, i.e., the C–N–C 𝒢1 and 𝒢2 molecular fragments, with the entire molecule **2** were analyzed (see entries 14 and 15 in Table 2). We noted that the environment around the N-atoms attracts **2** comparably with a slight preference of about −4.0 kcal mol^−1^ for RP-C3 for which we obtained a quantified attraction of −47.5 kcal mol^−1^ between 𝒢2 and **2**.

Let us now consider the four-atom fragments ℱ1 to ℱ4, which describe the immediate environment around the C-atoms of 𝒫 where the new C–C bonds with C28 are expected to form. Strong attractive interactions (−64.0 ± 4 kcal mol^−1^ on average) towards **2** were observed with a slight preference for RP-C3 again for which an attractive interaction of −67.7 kcal mol^−1^ was obtained (entries 16–19 in Table 2).

From a standard analysis of the REPs, it can be seen that the energy of adducts (**3**) for all possible sites of substitution are highly comparable. This suggests that there is no hindrance at this stage of the reaction. Furthermore, the RP-C2 and RP-C3 appear to be highly comparable in terms of their REPs and this is fully supported by the analyses performed above. This suggests a possible competition for nucleophilic addition at C2 and C3.

#### 3.1.2. Transition States and C–C Bond Formation

The addition of the nucleophile to the various identified electrophilic sites is not a spontaneous process. Hence, we performed the inter-nuclear distance scans (by 0.1 Å per step) between C28 and the C-atom (Cn) of the four potential RPs. In the adducts → transition state process, the molecules continuously rearrange themselves in such a way that allows C28 to get into close proximity with Cn. The reaction energy profiles computed along the respective reaction coordinates along all RPs are shown in Appendix A. The energy-optimized electronic structures of the TSs (**4a–d**) along all the RPs are shown in Appendix A. As an example, the TS structure obtained for the RP-C3 is shown in Figure 4.

As expected, from an analysis of the vibrational frequencies of these structures, a single negative frequency corresponding to the newly forming C–C bond was observed. These were verified as true TSs by means of IRC calculations where we observed that they do indeed link adducts (**3**) and the expected intermediates (**5**). The IRC graphs for all RPs can be seen in Appendix A.

Considering **4b** as an example (the TS along RP-C3 shown in Figure 4), an interesting arrangement is seen between four atoms. The C3 atom is facing the on-coming C28 atom with which a new bond is to be formed. The attractive intermolecular interaction energy between C3 and C28 increased from −20.8 kcal mol^−1^ in **3b** to −107.1 kcal mol^−1^ in **4b**. In addition, N4 is facing Li, as they will form a new bond as well and, at the same time, Li is still strongly attracted to C28 of **2**. The C2–C3 and C3–N4 bond lengths elongated from 1.43 and 1.31 Å in **3b** to 1.50 and 1.38 Å in **4b**, respectively. Another important geometrical change observed is in the DA (N1,C2,C16,C17). As the distance between the C-atoms involved in the newly forming C–C bond decreases, the phenyl substituent at C2 rotates to accommodate the addition of the nucleophile at C3 thus changing the DA from 23.08° in **3b** to −5.53° in **4b**.

Our computational modelling of the structures at the TSs showed that the interaction between the diminishing Li–C28 bond was much weaker than the interaction between the developing C–C for all RPs. This might suggest that all RPs will lead to the successful formation of the respective intermediates. However, values computed for the interactions between large molecular fragments (see entries 1–7 in Table 2) show the following trend in strengths of inter-fragment interactions RP-C3 > RP-C2 >> RP-C5 ≈ RP-C10 that correlates perfectly well with energy barriers for TSs 4 in Figure 2. To this effect, the strongest interactions (most negative) were obtained for the RP-C3 that has the lowest energy barrier. The interactions for RP-C5 and RP-C10 are weakest and highly comparable and the energy barriers for these reaction pathways are largest (36.8 and 36.7 kcal mol^−1^ for nucleophilic addition at C5 and C10, respectively) and, in principle, unsurmountable. Moreover, it is seen in Table 1 that the Li···C28 interaction is weaker along RP-C2 and RP-C3 relative to RP-C5 and RP-C10. This observation suggests that it would be easier to break the Li–C28 bond in **3a** and **3b**. This finding substantiates the lower energy barriers observed for RP-C2 and RP-C3 in Figure 2 when the C2–C28 or C3–C28 bond is formed.

Typically, in an attempt to rationalize the high energy barriers observed along RP-C5 and RP-C10, one might also follow the general rules applicable to organic chemistry. To this effect a benzene would be treated as a cyclic and planar molecule with sp^2^-hybridized C-atoms that, in turn allows the π-electrons to be delocalized in the molecular orbitals above and below the plane of the ring making benzene very stable. Atoms C5 and C10 are part of the benzene moiety of **1**. Therefore, nucleophilic addition at any of these C-atoms would lead to the disruption of the delocalization of the π-electrons around the benzene ring with no way of reforming it.

It was of interest and importance to find out whether the REP-FAMSEC based interpretations can support this generally accepted claim. We decided to study the impact of nucleophilic addition on the covalent interactions in ℬ𝓃 and 𝒫 rings which make up the quinoxaline moiety of **1**. The summed covalent interaction energies of these molecular fragments are included in Table 3. Looking first at the strength of the covalent interactions of ℬ𝓃, we see that at the adduct stage, the values are very comparable along all RPs. However, the Δ1 values in Table 3 show that at the TS stage, these interactions strengthened by approximately −20 kcal mol^−1^ in RP-C2 and RP-C3. The opposite trend is observed in RP-C5 and RP-C10 as at the TS stage, the covalent interactions of ℬ𝓃 have weakened by approximately 40 kcal mol^−1^. Hence, the effective change is highly in favor of RP-C2 and RP-C3 (by −60 kcal mol^−1^) and this agrees well with the generally accepted interpretations.

Considering now the pyrazine ring 𝒫, we see that at the TS stage, the covalent interactions in 𝒫 are weakened along all RPs as Δ2 >> 0. Although this is so, it is important to note that this effect is more significant along RP-C2 and RP-C3 where the covalent interactions in 𝒫 are weakened by 106.6 and 100.3 kcal mol^−1^, respectively.

To gain more insight, we decided to analyze the combined effect since the two rings are joint. To do so, one must consider the covalent interaction of the entire molecular fragment 𝒬 (see relevant data in Table 3). It shows that the strength of these interactions is very comparable at the adduct stage. Although these interactions weaken across all RPs at the TS stage (see Δ3 values in Table 3), they have weakened most along RP-C5 and RP-C10. Therefore, it is evident that nucleophilic additions at C5 or C10 impact both ℬ𝓃 and 𝒫 negatively, while nucleophilic additions at C2 or C3 only have a negative impact on 𝒫. This finding further supports our earlier conclusion that the nucleophilic addition at C5 or C10 must be eliminated from further considerations.

The elimination of RP-C5 and RP-C10 leaves us with two potential RPs, namely RP-C2 and RP-C3 with highly comparable energy barriers. One can argue that this is a result of the very comparable net atomic charges observed for C2 and C3 in Figure 1. Therefore, in an attempt to identify the preferred RP, we study the interaction energies between selected fragments of **2** and a set of major fragments of **1**. Prior to doing so, it is important to note that in the nucleophilic addition step of the reaction being studied, there is one bond (Li–C28) breaking and two bonds (Li–N1/N4 and C2/C3–C28) forming. That is, at the TS stage, Li is somewhere in the 3D space of the molecular system between N1/N4 and C28. As such, in our analysis of interaction energies of selected molecular fragments, Li is treated as a separate entity and thus all its interactions at this stage are seen as intermolecular interactions either with **1** (and its fragments) or with 𝓡.

From the data in Appendix A as well as entries 11, 22, and 23 in Table 2, it is clear that the Li-atom does not differentiate between the two RPs as its interactions with significant fragments of **1** and **2** are mostly comparable. The comparability of this data and the energy barriers suggests that the reaction has the potential to follow either RP-C2 or RP-C3, which would result in the formation of intermediates **5a** and **5b**, respectively. The formation of both **5a** and **5b** stabilizes the reaction (see Figure 2). One must note, however, the energy of **5a** is 1.1 kcal mol^−1^ above the initial reactants, whereas **5b** lies −4.9 kcal mol^−1^ below them. That is, **5b** is more stable, by 6.0 kcal mol^−1^, than **5a**.

The energy-optimized intermediates **5a** and **5b** are shown in Appendix A. Analysis of these electronic structures revealed that the elongated bonds C2–C3 and C3–N4, which characterized the **4b** TS along RP-C3, are elongated even further upon the formation of **5b**. The same phenomenon is observed for bonds N1–C2 and C2–C3 in the **4a**
→
**5a** process. Furthermore, upon the formation of **5a** and **5b**, the C-atom at which the nucleophile was added appears to have taken up a pseudo-tetrahedral (sp^3^) geometry, losing its original trigonal planar (sp^2^) geometry.

#### 3.1.3. Thermodynamic Analysis of the Nucleophilic Addition Step

To gain a comprehensive understanding of the assumed ONSH reaction mechanism followed by the reaction between **1** and **2**, one needs to investigate the changes in the thermodynamic parameters associated with the nucleophilic step as it is said to be the rate determining step (RDS) [34]. One of the most important experimental conditions associated with reactions involving organolithium reagents is to work at very low temperatures (195.15 K) as these reactions are highly exothermic. Therefore, all the changes in the thermodynamic properties were computed at 195.15 K. The data summarized in Table 4 reveal that upon the formation of adducts **3a** and **3b**, the *G* of the system increases slightly (by 1.8 kcal mol^−1^) along RP-C2 and decreases (by −2.6 kcal mol^−1^) along RP-C3. We also observe that *H* decreases along both RPs but in favor of RP-C3. As the reaction proceeds through the TS to the intermediates, both *H* and *G* follow the same trend. Although, relative to RP-C3, the increase in *H* and *G* is more significant by a few kcal mol^−1^ along RP-C2, it is evident that this does not exclude the RP-2.

From these observations, it follows that RP-C3 is both kinetically and thermodynamically more favorable than RP-C2. Although this correlates very well with the reported experimental work, from the evidence provided, one can argue that should the reaction have enough energy to overcome the energy barrier observed along RP-C2, intermediate **5a** could be formed and proceed with the subsequent hydrolysis step of the ONSH reaction mechanism. Hence, the hydrolysis of both **5a** and **5b** was modelled in the section that follows.

### 3.2. Hydrolysis

Hydrolysis is the second step of the ONSH reaction mechanism. It is important to note here that in the experimental work, water (H_2_O) was not explicitly introduced into the system. Rather, the system was exposed to the atmosphere and the moisture from the air served the purpose of hydrolyzing the intermediate formed at the nucleophilic addition step.

The hydrolysis step is initiated by the spontaneous (**∆***G* ~ −15 kcal mol^−1^) formation of adducts **6a**/**6b** (Figure 2) between the intermediates **5a**/**5b** and a water molecule **H_2_O** (see Figure 5). Importantly, all energy terms, *E*_ZPVE_, *G,* and *H* computed for **6a** and **6b** are lower (more negative) when compared with relevant data obtained for the initial reactants **1** and **2** (the full set of data is included in Table 5). Such significant stabilization of **6a** and **6b** must be attributed to the huge affinity between Li and O-atom of water due to the difference between their net atomic charges (−1.1385*e* for O-atom and nearly +1*e* for Li in both **5a** and **5b**). This was well revealed by our IQA and REP-FAMSEC analysis where the strongest attractive intermolecular interaction (−191.0 kcal mol^−1^) was observed between Li and O along both RPs. The overall interaction energies between the water molecule with either **5a** or **5b** were found to be very comparable (−60 kcal mol^−1^).

According to the general reaction scheme of the ONSH shown in Scheme 1, the hydrolysis process proceeds via a lithium-proton (Li–H) exchange reaction. That is, a proton from H_2_O is transferred to the N-atom in exchange for the Li-atom. This results in the formation of an amine at the N-atom and LiOH as a by-product.

To complete the modelling of this process the inter-nuclear distance between one of the protons from H_2_O and N1 (for RP-C2) or N4 (for RP-C3) was scanned in −0.1 Å steps using **6a** or **6b** as the respective input structures (see Appendix A for details). The energy-optimized electronic structures of the TSs **7a** and **7b** and hydrolysis products **8a** and **8b** are shown in Figure 6. Remarkably, the reaction energy profiles computed for the RP-C2 and RP-C3 are nearly identical when changes in the energy terms from **5** to **8** are considered (see Figure 2). That is, upon the exposure to moisture, the reaction can proceed via two, nearly spontaneous, parallel RPs. The energy barriers of these RPs at the TSs (**7**) are entirely negligible (about 1 kcal mol^−1^ for the free energy change and even smaller for *E*_ZPVE_ and *H*). However, only RP-C3 makes the provision for the aromaticity of the quinoxaline moiety to be recovered. That is, only **8b** can proceed with the final oxidation step of the ONSH reaction mechanism, whereas **8a** is trapped leading to a possible waste from an unwanted but very possible RP that will affect the yield of the reaction.

In order to further explore the relatively low yield for this reaction, we decided to investigate a possibility of another path for the RP-C2, even though most organic synthetic chemists would hardly consider this. We decided to test if, in principle, it is possible for H42 of water in **5a** to be transferred to C16 of 𝒫𝒽1. Relevant data obtained for the reaction energy profile along the H42···C16 reaction coordinates is shown in Appendix A, whereas the energies computed are included in Table 5. This new pathway produced the substitution of the phenyl at C2 to form (2-(2-phenylethynyl)quinoxaline with benzene and LiOH as by-products (**8e**) via the TS **7e** shown in Figure 6. Unexpectedly, the energy barrier (24.7 kcal mol^−1^, Table 5) at **7e** is entirely feasible for the reaction to proceed but not favorable when compared with that obtained for **7a/b**. This means that the Li-H exchange reaction leading to the formation of either **8a** or **8b** is kinetically more favorable than the transfer of H42 to C16. However, the product (**8e**) of this path is thermodynamically more stable. This is yet another possible waste of reactants that may contribute to the relatively low yield of the desired product.

## 4. Conclusions

Using the recently implemented REP-FAMSEC method [19], we quantified the energy contributions of all inter- and intramolecular interactions and their changes between consecutive steps of the assumed ONSH reaction mechanism for the reaction between 2-phenylquinoxaline **1** and lithium phenylacetylide **2**. The energies obtained for the selected molecular fragments that were driving a chemical change (they varied from a single atom to an entire molecule) explained the reaction energy profiles computed for four potential sites of the nucleophilic addition.

According to the experimental report by Ndlovu et al. [16], this reaction occurs at the position occupied by a hydrogen atom in the electron deficient ring (C3 in **1**) to form the so called σ^H^-adduct. However, from the generated molecular graph of **1**, three other sites, namely C2, C5, and C10, were identified as potential competing sites for the reaction being studied. This suggested that the nucleophilic addition, which is said to be the key and rate determining step of the ONSH reaction mechanism, could occur at various sites, resulting in multiple reaction pathways (RPs) and, consequently, affect the yield of the desired product.

The REP-FAMSEC based study established that the two reactants have a strong affinity for each other driven by the strong attractive interactions between the Li-atom and either N1 or N4. From the two unique approaches between **1** and **2**, four different adducts (**3**), each well prepared for the subsequent formation of a C–C bond at either one of the four identified sites, were formed. These were very comparable in energy and all lead to the stabilization of the reaction. However, as the reaction proceeded through to the TS (**4**) to form the intermediates (**5**), very high-energy barriers were observed for RP-C5 and RP-C10. Supported by the computed changes in the integrity of covalent interaction in molecular fragments ℬ𝓃, 𝒫, and 𝒬 as well as our understanding of general/organic chemistry, these RPs were concluded to be impractical or impossible and therefore eliminated as possible RPs.

Following the proposal made by classical chemists that steric hindrance could be the contributing factor towards the relatively low yield (47%) obtained for this reaction, we computed the interactions between the phenyl substituent (𝒫𝒽1) at C2 and either the entire molecule **2** or fragments of it. From this data, we observed that the phenyl substituent guided the nucleophile towards RP-C2 and RP-C3, indicating that the preferred RP cannot be attributed to the steric hindrance caused by 𝒫𝒽1.

Although the data obtained at the nucleophilic addition stage indicated that RP-C3 was both kinetically and thermodynamically favored over RP-C2, the energy barriers observed at this stage were very comparable for both RPs. This indicates that both RPs (C2 and C3) can progress to form intermediates **5a** and **5b**, which will undergo hydrolysis (the second step of the ONSH reaction mechanism). Upon the introduction of H_2_O to the system, we observed that both RPs were nearly spontaneous towards their respective hydrolysis products (**8a** and **8b**). However, of the two products, only **8b** can proceed to the final oxidation stage of the ONSH reaction mechanism to give the expected 2,3-disubstituted quinoxaline compound. This clearly points out that there exists an unwanted yet very possible RP (RP-C2) leading to a possible waste (**8a**) that may justify the yield obtained for the desired product.

Furthermore, a secondary RP along RP-C2 where H42 from H_2_O in **6a** is transferred to C16 rather than N1 was found. Although this path had the highest energy barrier at the hydrolysis stage, it was not high enough to be eliminated as a real possibility. This RP led to the irreversible formation of 2-(2-phenylethynyl)quinoxaline with lithium hydroxide and benzene as by-products (**8e**). This points out towards yet another very possible RP leading to an unwanted output from the hydrolysis stage that will lead to low yields for the desired product.

Clearly, RP-C2 competes with RP-C3 leading to a possible mixture of the respective products. In order to eliminate RP-C2 or to minimize its influence on the yield for the desired product, we strongly recommend that experimental chemists run the first stage of the reaction at a low temperature for a longer time (2–4 h) in a moisture-free and oxygen-free environment (glove box). The extension of time will allow an equilibrium to be reached with **5b** as the predominant intermediate. That is, even though the reaction will follow both RP-C2 and RP-C3 initially, having sufficient time and assuming the spontaneous drive of a reaction environment towards the lowest energy possible, **5a** should go back to reform adducts, particularly **3b**, and follow RP-C3 towards **5b**. In addition, the in situ preparation of the Li-nucleophile should be done with precaution to ensure that a sufficient amount is present during the reaction.

## Data Availability

On request, computational data is available from IC.

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
