# Peer review of "A REP-FAMSEC Method as a Tool in Explaining Reaction Mechanisms: A Nucleophilic Substitution of 2-Phenylquinoxaline as a DFT Case Study"

_molecules, 2021, doi:10.3390/molecules26061570_

Round 1
Reviewer 1 Report
This manuscript dedicated to systematical theoretical consideration of reaction 2-phenylquinoxaline with nucleophiles.
The authors reported interesting results, which are suited for Molecules readership. However, the presentation of results must be improved.
Namely, the authors should make one general scheme and show all discussed compounds and intermediates: 1, 2, 3, 4, 5, 6, 7, 8.
Additionally, some employed computation approaches are debate question. Particularly, the significance of QTAIM for chemistry is increasingly being questioned: Perrin, JACS, 1991, 113, 2865; Grimme et al, Angew. Chem. Int. Ed. 2009, 48, 2592; Clark et al, Phys. Chem. Chem. Phys. 2018, 20, 30076. I suggest that this should be mentioned.
Author Response
We do not agree that a general scheme showing all the compounds discussed in the manuscript is necessary for this paper. This we feel will make the paper difficult to follow as some of the compounds i.e. 3, 4 and 6 are either adducts or intermediates, which have four possible conformations. Fitting all these structures into one scheme will take up too much space and create confusion. In addition, such as scheme is best placed at either the introduction section or conclusion section. We don’t feel it is wise to put the scheme in the introduction section as most of the structures presented in this paper were only discovered after our research and not through generalised chemistry theory. If the scheme is placed on the conclusion section, we feel that this will be a repetition of data that has already been reported in the discussion section.
We also do not agree with the reviewer’s statement that ‘ the significance of QTAIM for chemistry is increasingly being questioned:…”.
Firstly, we are not aware of a single paper that questions Bader’s energy partitioning scheme called QTAIM – please let us know about any. To this effect, let us cite a statement made in the second paper indicated by the Reviewer 1 - Grimme et al, Angew. Chem. Int. Ed. 2009, 48, 2592. Please see the first paragraph of the right-hand column on page 1: ‘Note that we question neither the foundations of AIM nor its usefulness for the interpretation of electronic structure.’
In the following paragraph in the same column, a real controversy between an orthodox approach and Bader’s interpretation is explained, namely:
‘The existence of a BCP and a BP is established as a necessary and sufficient condition for a bond in the basic interpretation of AIM.[3,4] In more recent literature this concept has been extended even further, and BCP and BP are often related to attractive interactions (see, for example, Ref. [7]). The most controversial aspect of this interpretation is the concept of H–H bonding, which was introduced by Matta and co-workers.’
Clearly, this is not QTAIM that is being questioned but the interpretation of QTAIM-based interpretation of a chemical bonding. And this is not surprising because Bader’s suggestion hit the wall of an orthodox interpretation of chemical bonding that originates from the Lewis concept of electron-pair shared by the atoms, an over a 100-years old concept that chemists found difficult to depart from till this day.
A very important point must be stressed here, namely that as pointed out by Grimme et al (see the first paragraph of this paper):
‘The concept of chemical bonding is fundamental to natural science. [1] Despite its importance, a precise and unambiguous definition of when a chemical bond exists between (usually two) atoms is difficult. … The other approach is based entirely on theory in the context of ab initio schemes derived from wave function and electron density calculations. Although these methods have advanced extraordinarily in recent years, [2] the theoretical definition of bonding is still problematic. The simple reason is that no quantum mechanical “bond operator” exists that would provide the desired answer, for example, as a conventional expectation value.’
Clearly, this is not the QTAIM but a concept of chemical bonding steamed from QTAIM that became controversial. Importantly, our manuscript is not about defining chemical bonding at all. Our work is on a reaction mechanism and this is why we disagree with the suggestion made by Reviewer 1, namely:
‘: Perrin, JACS, 1991, 113, 2865; Grimme et al, Angew. Chem. Int. Ed. 2009, 48, 2592; Clark et al, Phys. Chem. Chem. Phys. 2018, 20, 30076. I suggest that this should be mentioned.’
In our opinion this does not fit our contribution at all.
Finally, we would like the reviewer 1 to read our most recent paper (T.G. Bates et al, J. Comput. Chem, 2021 https://doi.org/10.1002/jcc.26491) that brings new light on the CH…HC interactions.
Reviewer 2 Report
In this work, Cukrowski et al. employed a computational approach to provide insight into a reaction mechanism of a nucleophilic substitution of 2-phenylquinoxaline. This aromatic heterocyclic organic compound play an important role in the Oxidative Nucleophilic Substitution of Hydrogen (ONSH) reaction, so the work fits the current trends in chemistry well. The work presents very interesting research results. It is also well written. The calculations have been done carefully and no obvious errors or omissions could be detected by referee. Therefore, I can recommend publication after minor revision.
The primary step of investigated reaction is determined mainly by the location of the nucleophilic/electrophilic centers in the molecule. The authors used the QTAIM-defined net atomic charges to determine the location of nucleophilic center. Is it also possible to locate the fragment of the molecule sensitive to an electrophilic/nucleophilic substitution using the frontier orbital theory? Which of these methods will be more appropriate to locate the nucleophilic/electrophilic centers of molecule?
Author Response
Thank you very much for highly positive feedback and positive recommendation.
Considering two questions posed by the Reviewer: “Is it also possible to locate the fragment of the molecule sensitive to an electrophilic/nucleophilic substitution using the frontier orbital theory? Which of these methods will be more appropriate to locate the nucleophilic/electrophilic centers of molecule?“
Our immediate answer is:
We do not know, as this would have to be studied extensively – one molecular system cannot provide definite answers. However, we are grateful for this suggestion and will examine the applicability of the ‘frontier molecular orbital theory’ in near future. Kenichi Fukui has spent a lot of time (years) and effort in connecting chemical reactivity with the HOMO-LUMO interactions (he shared the Nobel Prize in Chemistry with Roald Hoffmann for his work on reaction mechanisms). The reviewer’s suggestion is perfectly aligned with our recent interest on interpreting Molecular Orbitals in terms of their contribution to chemical bonding (hence most likely also chemical reactivity) and our first reports show that one must be extremely careful in drawing conclusive remarks when interpreting MOs contributions to an inter-nuclear region (where bonding, interaction reactivity takes place) is considered.
To this effect, see:
1) J. H. de Lange et al, Phys. Chem. Chem. Phys., 2019, 21, 20988—20998;
2) I. Cukrowski et al, J. Phys. Chem. A 2020, 124, 5523−5533;
3) S. de Beer, J Comput. Chem. 2020, 41, 2695–2706;
4) T.G. Bates et al, J. Comput. Chem, 2021 https://doi.org/10.1002/jcc.26491)
Finally, the QTAIM charges we used are just a first pointer in identifying potential nucleophilic/electrophilic centres. Our REP-FAMSEC based analysis makes an extensive use of the IQA-defined energy terms; the latter are incorporated in the energy terms defined in the REP-FAMSEC methodology. As far as we know, the REP-FAMSEC method is the only one that can freely focus on a selected atom, atom-pair or a selected molecular fragment and analyse their impact on the progress of a reaction. It is quite clear that Molecular Orbitals that are, by definition, molecular-wide, are not able to provide such focused information.
Reviewer 3 Report
The manuscript “A REP-FAMSEC Method as a Tool in Explaining Reaction Mechanisms: A Nucleophilic Substitution of 2-Phenylquinoxaline as a DFT Case Study” by I. Cukrowski and coworkers deals with a detailed analysis of the mechanism of a reaction between the title compound and lithium phenylacetylide. The main objective of the manuscript is unraveling the reason behind the reaction’s low yield. The calculations were based on the REP-FAMSEC (Reaction energy profile and fragment attributed molecular system energy change) methodology. Four reaction pathways were investigated and the most favorable ones identified. Moreover, the authors suggested which experimental conditions should be used in order to increase the yield of the desired product. The manuscript is clearly and methodically written and would be of interest both to computational and organic chemists. I would only ask the authors to consider rephrasing the sentence on line 311: “The success of many organic reactions depends on the structure of the TS. This structure must resemble the forming intermediate more than the reactants.” This statement is valid provided the intermediate is less stable than the reactants, which is the case in the studied reactions. If that were not the case, the transition state would resemble the adduct.
Author Response
Thank you very much for highly positive feedback. Regarding the Reviewer’s request ‘I would only ask the authors to consider rephrasing the sentence on line 311: “The success of many organic reactions depends on the structure of the TS. … ‘
We are not entirely sure regarding this suggestion. Hence, to avoid any controversy we changed the beginning of this paragraph:
‘The success of many organic reactions depends on the structure of the TS. This structure must resemble the forming intermediate more than the reactants. This was confirmed by our computational modelling where the interaction between the diminishing Li–C28 bond was observed to be much weaker than the interaction between developing C–C for all RPs. This might suggest that all RPs will lead to the successful formation of the respective intermediates.’
To
‘Our computational modelling on the structures at the TSs showed that the interaction between the diminishing Li–C28 bond was much weaker than the interaction between developing C–C for all RPs. This might suggest that all RPs will lead to the successful formation of the respective intermediates.’